# Alternative Splicing: Expanding the Landscape of Cancer Biomarkers and Therapeutics

**DOI:** 10.3390/ijms21239032

**Published:** 2020-11-27

**Authors:** Cláudia Bessa, Paulo Matos, Peter Jordan, Vânia Gonçalves

**Affiliations:** 1Department of Human Genetics, National Health Institute Dr. Ricardo Jorge, 1649-016 Lisbon, Portugal; claudia.bessa@insa.min-saude.pt (C.B.); paulo.matos@insa.min-saude.pt (P.M.); 2BioISI—Biosystems & Integrative Sciences Institute, Faculty of Sciences, University of Lisbon, 1749-016 Lisbon, Portugal

**Keywords:** pre-messenger RNA, alternative splicing, mutation, cancer progression, splicing factor, signal transduction, biomarker, therapeutic target, tumor biology

## Abstract

Alternative splicing (AS) is a critical post-transcriptional regulatory mechanism used by more than 95% of transcribed human genes and responsible for structural transcript variation and proteome diversity. In the past decade, genome-wide transcriptome sequencing has revealed that AS is tightly regulated in a tissue- and developmental stage-specific manner, and also frequently dysregulated in multiple human cancer types. It is currently recognized that splicing defects, including genetic alterations in the spliced gene, altered expression of both core components or regulators of the precursor messenger RNA (pre-mRNA) splicing machinery, or both, are major drivers of tumorigenesis. Hence, in this review we provide an overview of our current understanding of splicing alterations in cancer, and emphasize the need to further explore the cancer-specific splicing programs in order to obtain new insights in oncology. Furthermore, we also discuss the recent advances in the identification of dysregulated splicing signatures on a genome-wide scale and their potential use as biomarkers. Finally, we highlight the therapeutic opportunities arising from dysregulated splicing and summarize the current approaches to therapeutically target AS in cancer.

## 1. Introduction

In higher eukaryotes, the primary gene transcripts, also called precursor messenger RNAs (pre-mRNAs), undergo a finely tuned post-transcriptional regulatory process that removes the non-coding regions (introns) and splices together the coding sequences (exons), thus generating the mature mRNAs. This mechanism is designated as pre-mRNA splicing and is a critical step in gene expression. In addition, it is well known that the splicing patterns of a gene vary widely as result of the process of alternative splicing (AS) that differentially retains or excludes certain exons from the pre-mRNA transcript. Consequently, various combinations of exons from a single gene can produce a diversity of mRNA variants, which is determinant to structural transcript variation and proteome diversity [1] and can generate different protein isoforms with related, distinct, or even opposing functions [2,3]. Remarkably, AS is a widespread event affecting more than 95% of transcribed human genes, as suggested by data provided by whole transcriptome sequencing projects [2,4]. This complex and tightly regulated mechanism is shared across different tissues and developmental stages, and frequently dysregulated in various human diseases, including cancer [5]. This dysregulation was verified in various types of cancer through detection of aberrant splicing patterns in tumor tissues when compared to their normal counterparts by high-throughput sequencing techniques [6,7,8,9]. Additionally, accumulating evidence clearly supports that the aberrant splicing profiles found in cancer are contributing to neoplastic transformation, cancer progression, and therapy resistance [10,11]. Therefore, it is of utmost relevance to identify pathological splicing isoforms for the development of new effective biomarkers, as well as to clarify the mechanisms behind aberrant AS, thereby elucidating its impact on cancer and providing novel therapeutic strategies.

Hence, this review summarizes our current understanding of splicing alterations in cancer and emphasizes the need for a deeper understanding of cancer-specific splicing programs in order to provide new insights in oncology. Particularly, we highlight the relevance of identifying cancer-specific AS events for the development of novel biomarkers and discuss part of the current therapeutic landscape regarding splicing-based therapies for cancer treatment.

## 2. Alternative RNA Splicing: An Overview

Pre-mRNA splicing consists of a multistep process orchestrated by the spliceosome, a huge RNA/protein complex comprising five small nuclear ribonucleoproteins (snRNPs; U1, U2, U4, U5, and U6) and numerous associated proteins [12,13]. Briefly, the reaction initiates with the assembly of an initial spliceosome complex through recognition of critical consensus splice sites at the pre-mRNA transcript, as schematically represented in Figure 1A. It comprises a stepwise process that begins with the recruitment of U1 snRNP to the 5′ splice site. Then, the splicing factor 1 (SF1), U2 snRNP auxiliary factor 2 (U2AF2), and U2 snRNP auxiliary factor (U2AF) 2, and U2AF 1recognize the branch point site (BPS), the polypyrimidine (poly-Y) tract, and the AG dinucleotide of the 3′ splice site region, respectively. The occupancy of these three consensus sequences induces the association of U2 snRNP with the BPS, which is further stabilized by the U2 snRNP component SF3B1. Consequently, intronic recognition prompts the engagement of U4/U6/U5 tri-snRNP with the complex, and subsequent formation of a catalytically inactive complex. This leads to several conformational and compositional rearrangements of spliceosomal components, including the dissociation of U1 and U4 snRNPs, which in turn promotes the formation of the activated spliceosome that catalyzes the splicing reaction [14]. Transcripts from nearly all protein-coding genes undergo one or more types of AS, giving rise to different mRNAs that differ in transcript degradation or are translated into alternative protein isoforms in a cell type-, organ-, or tissue-specific manner [2,4,15]. In higher eukaryotes, among the currently known AS events represented in Figure 1B, the most common is exon skipping [16], accounting for approximately 40% of all AS events, in which a cassette exon is removed from the pre-mRNA together with its flaking introns. Besides this, switching between alternative 5′ and 3′ splice site positions, mutually exclusive splicing of adjacent exons and differential retention of introns are also important variations of AS (Figure 1B). Other types of AS events include the use of alternative transcription start sites and alternative polyadenylation.

In AS, the regulated process consists of the recognition of an exon by the spliceosome. For this, splice site utilization is further regulated by cis-acting splicing-regulatory elements, which either promote or inhibit the use of adjacent splice sites by recruiting trans-acting splicing factors [17]. Thus, they are classified into exonic or intronic splicing enhancers (ESE/ISE) or silencers (ESS/ISS), depending on their positions and functions (Figure 1C). In general, enhancers are recognized by trans-acting factors belonging to the serine/arginine-rich (SR) protein family to facilitate splice site recognition and exon inclusion [18]. On the other hand, silencers usually interact with other types of trans-acting factors such as heterogeneous ribonucleoproteins (hnRNPs) to inhibit splice site recognition and promote exon skipping [2]. However, several AS events exist in which SR or hnRNP proteins act as inhibitors or enhancers of splicing, respectively.

### 2.1. Dysregulation of Alternative Splicing in Cancer

Cancer mainly evolves through successive genetic alterations and genomic dysregulation, but is also affected by the tumor microenvironment. These render oncogenes constitutively active and inactivate tumor-suppressor genes. As a result, cancer cells acquire specific abilities during tumor development, including self-sufficiency in growth signals, insensitivity to growth inhibitory signals, evasion of apoptosis, limitless replicative potential, sustained angiogenesis, and tissue invasion and metastasis [19]. These processes can also be dysregulated by AS, which in turn can generate variant proteins with altered physiological function [3]. Particularly, a recent systematic study performed by Kahles et al. reported that AS events are more frequent in cancer tissues compared to normal ones, and many of them are cancer-type specific [20]. Among the factors that can trigger aberrant AS, somatic mutations that disrupt splicing regulatory motifs, as well as mutations or expression changes in components of the core splicing machinery or splicing auxiliary factors, are frequently described [6,7,21,22,23,24].

Aberrant splicing in cancer has been widely linked to mutations creating cis-regulatory motifs that generate novel splice sites, as demonstrated by the discovery of almost 2000 splice site-creating mutations through a robust whole-exome analysis encompassing more than 8000 tumor samples across 33 cancer types [25]. One of the AS events frequently associated with these somatic mutations is intron retention, and mainly affects tumor suppressor genes such as *TP53*, *ARID1A*, and *PTEN* [7]. Importantly, most of the intron retention events are able to induce frameshifts in pre-mRNA sequence, resulting in the generation of premature termination codons (PTCs), which in turn leads to the degradation of the transcript through nonsense-mediated mRNA decay (NMD) or to the production of truncated proteins (e.g., dominant negative isoforms or neo-antigens). Interestingly, somatic exonic mutations have also been reported in oncogenes, particularly in ESE and ESS sequences [6], and associated with the generation of pro-tumorigenic variants.

Recurrent somatic mutations affecting the components of the early spliceosome complex formation have frequently been described in cancer, particularly in hematological malignancies, including myelodysplastic syndromes (MDS), other myeloid neoplasms, and chronic lymphocytic leukemia (CLL) [26,27,28]. Among the genes most affected by these mutations that almost always occur in a mutually exclusive manner are *SF3B1* (splicing factor 3b subunit 1), *SRSF2* (serine/arginine-rich splicing factor 2), *U2AF1* (U2 small nuclear RNA auxiliary factor 1), and *ZRSR2* (zinc finger RNA binding motif and serine/arginine rich 2) [26]. SF3B1, a subunit of the U2 snRNP that recognizes the BPS, is the most commonly mutated splicing regulator in numerous cancers, with a prevalence ranging from 5% in breast cancer to 81% in an MDS subtype [29]. Cancer-associated *SF3B1* mutations are located within HEAT (Huntingtin, Elongation factor 3, protein phosphatase 2A, Targets of rapamycin 1) domains, which are involved in protein–protein interactions and clustered in hotspots, namely K700, E622, R625, H662, and K666. Specifically, they are mainly related with the binding of SF3B1 to cryptic 3′ splice sites, located in regions with shorter and weaker poli-Y tracts, and consequently linked to aberrant BPS usage [22,30,31]. This abnormal assembly of spliceosome originates many mRNAs with a PTC, which are subsequently degraded by NMD.

Although the mechanism that induces the change of 3′ splice site usage by SF3B1 is not fully elucidated, it is hypothesized that these mutations alter the interaction of SF3B1 with other spliceosomal components required for BPS recognition. SRSF2 is a member of the SR protein family that binds to specific ESE sequences, namely CCNG or GGNG, through its RNA recognition motif (RRM) domain, and recruits U1 snRNP and U2AF to the 5′ and 3′ flanking splice sites, respectively [32]. This splicing regulator has also been found recurrently mutated, particularly in patients with MDS and chronic myelomonocytic leukemia (CMML) [26]. SRSF2 mutations predominantly occur at the P95 residue, which is located near the RRM domain [26]. According to several reports, these mutations change the RNA-binding affinity of SRSF2, favoring the recognition of C-rich CCNG over G-rich GGNG motifs in ESE consensus sites, which in turn leads to misregulation of exon inclusion [33,34]. The gene encoding UA2F1 is also mutated in myeloid malignancies, as well as in lung adenocarcinomas [26,35,36]. U2AF1 hotspot mutations occur almost exclusively at S34 and Q157 residues within the two conserved zinc-finger domains, thus affecting the recognition of the 3′ splice site AG motif [37,38]. In contrast to mutually exclusive hotspot mutations described for *SF3B1*, *SRSF2*, and *U2AF1*, *ZRSR2* mutations are distributed throughout the gene and most are consistent with a loss-of-function phenotype [23]. In 2015, in addition to the major (or U2) spliceosome, ZRSR2 was also characterized as an essential component of the minor (or U12) spliceosome that catalyzes the processing of a distinct class of introns (U12-type introns). Particularly, it is involved in 3′ splice site recognition in U12 snRNA-dependent splicing, so that mutations in this gene are associated with an increase in the retention of U12-type introns [23].

Apart from genomic mutations, the pre-mRNA splicing of many genes related to cancer pathogenesis can also be disturbed by changes of the copy number or expression levels of splicing factors [39]. Actually, abnormal expression of several splicing factors have frequently been reported in solid tumors and closely associated with cancer development and progression, even in the absence of mutations [40,41,42,43]. One of the best characterized is the serine-arginine splicing factor 1 (SRSF1; formerly known as ASF or SF2), an SR protein involved in both constitutive and AS, as well as in other cellular processes. It is upregulated in several human tumors, including colon, breast, thyroid, small intestine, kidney, and lung, and its experimentally induced overexpression leads to the transformation of human and mouse mammary epithelial cells, suggesting that it acts as a proto-oncogene [44,45,46]. Until now, SRSF1 upregulation has been shown to affect many AS events in cancer-associated genes. In particular, SRSF1 overexpression induces an increase in the levels of oncogenic protein isoforms of RON [47], MNK2, and S6K1 [44] and of the anti-apoptotic isoforms Bcl-xL and MCL-1L [48], and a loss of the tumor suppressor isoform of BIN1 [44]. Curiously, the overexpression of hnRNP A1 and hnRNP A2/B1, two factors previously suggested to antagonize SR proteins, was also reported in lung, breast, and brain tumors [49,50,51,52]. Interestingly, in glioblastoma (GBM) cells, hnRNP A2/B1 showed splicing effects similar to the proto-oncogenic SR protein SRSF1 [52]. More recently, hnRNP A2 (as well as B1 and K) has been associated with enhanced expression of anti-apoptotic variants of BIN1 and CASP9, and decreased expression of the pro-apoptotic variant Bcl-xS [48], promoting the same phenotypic response as SRSF1 overexpression.

The major drivers of aberrant splicing profiles appear to be changes in the expression levels of splicing factors; however, the mechanisms behind the altered expression of the splicing factors in tumors are not yet fully understood. Although sporadic somatic mutations in genes encoding splicing factors have already been recurrently detected in solid tumors [43], it is widely recognized that oncogenic signaling has a central role [53]. Actually, abnormal activation of signaling pathways has been extensively reported in cancer. For instance, in colon cancer, oncogenic Kirsten rat sarcoma viral (KRAS) activates the RAS–MAPK pathway, leading to an increase in the expression levels of the AS factor polypyrimidine tract-binding protein 1 (PTBP1), activated via transcription factor ELK1. In turn, increased PTBP1 levels induce a shift in the AS of tumor-associated transcripts, namely, the small GTPase Ras-related C3 botulinum toxin substrate 1 (RAC1), adaptor protein NUMB, and PKM [54]. In addition to transcriptional stimulation of PTBP1 downstream of RAS, ERK was reported to phosphorylate the splicing factor SAM68, thereby inducing the binding of phospho-SAM68 to the 3′UTR of the *SRSF1* transcript [55]. This binding promotes the retention of an intron required for production of full-length SRSF1 and prevents the downregulation of *SRSF1* transcripts through the NMD pathway. Consequently, the increased SRSF1 levels, comparable in effect to the above described *SRSF1* gene amplifications [44], induce a switch in AS of the *RON* gene transcripts, favoring the production of the oncogenic isoform RONΔex11. Phosphorylated SAM68 further stimulates inclusion of the variable exon 5 sequence into the CD44 mRNA, generating a pro-invasive cell adhesion protein variant [56].

Another MAPK pathway responds when cells experience physiologic stress. Osmotic stress triggers the MKK(3/6)-signaling cascade, leading to p38-activation, which upon nuclear translocation induces hnRNP A1 phosphorylation, followed by its export into the cytoplasm [57,58]. The corresponding decrease in nuclear splicing factor abundance is sufficient to change AS patterns. The PI3K/AKT signaling is another key pathway involved in cell survival and escape from apoptosis in numerous solid tumors. In non-small cell lung cancers (NSCLC), it was demonstrated that the activation of the PI3K/AKT pathway by oncogenic factors mediates the exclusion of the exon 3,4,5,6 cassette of CASP9 transcripts’ via the phosphorylation state of SRSF1, thus generating the anti-apoptotic Casp-9b isoform [59]. At the same time, AKT-mediated phosphorylation of hnRNPL induces its binding to a splice silencer element in Casp-9 pre-mRNA, further enhancing the exclusion of the exon cassette [60,61]. AKT activation also leads to phosphorylation and nuclear translocation of SR proteins, causing alternative exon inclusion in the fibronectin pre-mRNA [62]. Interestingly, in colorectal cells, inhibition of PI3K/AKT signaling led to increased expression of endogenous SRSF1, leading to the inclusion of an alternative exon, termed 3b, in the mRNA of the small GTPase RAC1, which generates the pro-tumorigenic splice variant RAC1B [63]. Later, it was described that SRPK1 and GSK3β act upstream of SRSF1, and are required to sustain RAC1B splicing in colorectal cancer (CRC) cells [64]. Particularly, it was shown that GSK3β indirectly regulates the levels of SRSF1 and RAC1B via SRPK1, since its depletion leads to a reduction of SRPK1 activity towards SRSF1, and a concomitant decrease in nuclear SRSF1 levels, resulting in less RAC1B generated. Another central hub of oncogenic signaling is the Wnt pathway, which is activated in many colorectal tumors. Remarkably, this pathway also modulates RAC1B splicing in CRC cells: It was described that the *SRSF3* gene encoding splicing factor SRSF3/SRp20 is a transcriptional target for activated β-catenin/TCF4 complexes, leading to increased SRSF3 protein levels [65]. In a subsequent work, it was demonstrated that increased *SRSF3* transcription following activation of the β-catenin/TCF4 pathway suppresses RAC1B splicing through SRSF3-mediated exclusion of exon 3b from the RAC1 mRNA [63]. Together, these examples show how signaling mechanisms affect alternative pre-mRNA splicing and change tumor-related gene expression.

### 2.2. Examples of Cancer-Associated Alternatively Spliced Variants

Several splice variants have been associated with different hallmarks of cancer, including initiation, progression, and metastasis. In Table 1, we highlight some of the most relevant AS events in cancer-associated genes involved in different steps of oncogenic transformation, as well as the types of cancer they are most often associated with. Other examples were listed in a recent review [66].

## 3. RNA Splice Variants as Potential Biomarkers in Cancer

Early detection and diagnosis of cancer as well as the identification of the most effective personalized therapy for each patient remain the main challenges in oncology. Over the past few years, cancer biomarkers have emerged as valuable screening, diagnostic, and prognostic tools, enabling us to classify the extent of disease, define the prognosis, select the most appropriate treatment regimens, or follow up on the clinical response after treatment or surgical intervention [113]. Despite the advances, the development of more efficient biomarkers is still needed. Indeed, the amount of candidate cancer biomarkers that have been approved for clinical practice is too low, indicating that the majority of them are poor predictors of disease and treatment outcome, and are thus not reliable clinical tools [114]. In order to fill this gap, the biomarker potential of AS in cancer is currently being explored. Notably, the technological developments in sequencing and bioinformatics have provided extensive information to identify AS targets on a genome-wide scale, and in turn pathways and cellular programs that are differentially regulated in cancer cells [115,116,117,118,119,120,121]. However, from this large-scale approach, hundreds of splicing alterations are obtained that result either from mutations or abnormal expression of splicing factors, but do not readily allow for the identification of the critical cancer-driving splicing events. On the other hand, although individual pathogenic splicing events have already been described, systematic studies of the functional impact of widespread splicing alterations in cancer have yet to be performed. Actually, it is crucial to determine the outcome induced by the observed splicing changes in a tumor-specific manner with corresponding resolution at the proteome level. Therefore, in order to overcome this issue of data science and explore the splicing opportunities in precision medicine, it is extremely important to use robust analysis methods able to predict and validate reliable cancer-associated splicing changes.

To date, several cancer-specific alternative transcripts with potential prognostic and predictive value in clinical settings were identified. For instance, the presence of the alternatively spliced androgen receptor variant 7 (*AR-V7*) in castration-resistant prostate cancer patients has been linked to a decrease in the effectiveness of hormone-directed therapy [122]. In pancreatic ductal adenocarcinoma patients receiving radical surgery and adjuvant chemotherapy, it was suggested that the tumors with higher basal expression of PKM2 exhibit more aggressive behavior and worse response to chemotherapy [123]. Additionally, EGFR variants have been widely reported in various tumor types and related to tumor progression [103,104,105,124,125]. In some cases, however, the evidence is less clear. For example, the prognostic value of the CD44 variant 6 (CD44v6) in CRC was debated for years due to contradictory results [126,127,128]. Nevertheless, further studies reinforced the relevance of CD44v6 as an independent negative prognostic factor and a promising therapeutic target in CRC [94,95,129]. Another example of a splicing biomarker with predictive potential in CRC is the upregulation of RAC1B. The overexpression of this RAC1 splice variant is frequent in CRCs carrying *BRAF^V600E^* mutation, which in advanced-stage tumors is a recognized poor prognostic biomarker [130]. Moreover, it was also reported that RAC1B expression impacts the clinical outcome of metastatic CRC patients treated with first-line 5-fluorouracil/leucovorin plus oxaliplatin or capecitabine plus oxaliplatin (FOLFOX/XELOX) chemotherapy. Indeed, the results obtained indicate that RAC1B overexpression represents an independent predictive marker of poor outcome in *KRAS/BRAF* wild-type metastatic CRC patients treated with this adjuvant therapy [131]. In 2013, the overexpression of RAC1B in papillary thyroid carcinomas (PTCs) was documented for the first time, and a possible interplay between BRAF*^V600E^* mutation and RAC1B postulated that may contribute to an unfavorable prognosis [132], which was also proposed for follicular thyroid carcinomas [101]. Later, the pro-tumorigenic advantage of RAC1B overexpression in thyroid carcinomas was linked to the induction of apoptosis resistance through NFκB activation [102]. Curiously, in CRC, RAC1B expression was also described as conferring chemoresistance to oxaliplatin through activation of NFκB signaling [133]. Despite being preliminary, these results indicate that RAC1B may be a clinically useful prognostic molecular biomarker for disease progression as well as a marker of resistance to therapy.

### Genome-Wide Identification of Cancer-Associated Splicing Signatures

Recent advances in high-throughput screening (HTS) technologies covering whole-genome and -exome sequencing, such as RNA sequencing (RNA-seq), have greatly contributed to improving the diagnosis and treatment of human diseases. Particularly, RNA-seq currently represents one of the most powerful tools to investigate AS at the genome-wide level [134,135,136,137,138,139]. Compared with gene expression microarrays, also designed to sample AS events on a genome scale, RNA-seq exhibits various potential advantages, including the ability of estimating the abundance of known and novel alternative transcripts, and providing better resolution, deeper coverage, and higher accuracy [139]. However, this technology still presents some drawbacks, namely the high cost of sequencing at deep coverage and the need to continually optimize the bioinformatics protocols for processing and analyzing RNA-seq data. On the other hand, most standard RNA-seq-based analyses have mainly focused on changes in gene expression level, thus lacking information about slight differences in alternative isoform usage and exon inclusion or exclusion. The importance of investigating AS profiles in RNA-seq data was recently highlighted by a study in a preclinical model of progressive diabetic nephropathy [135]. Using the isoform- and exon-level analysis of RNA-seq data, the authors identified AS patterns in genes implicated in disease pathogenesis, such as *SHC1*, *SERPINC1*, *EPB4.1L5*, and *IL-33*, which would have been overlooked by standard gene-level analysis.

Similarly, the profiling of AS signatures can be expected to provide insight into the disease process or identify potential prognostic or therapeutic biomarkers for cancer. Indeed, with the advent of HTS technologies, several studies have focused on detecting cancer-specific AS events by comparing cancer tissues with normal controls. In 2013, Eswaran et al. described for the first time splicing signatures specific to one of the three breast cancer sub-types. They further revealed that exon skipping and intron retention were the predominantly occurring splicing changes and also uncovered previously unknown isoforms of CDK4, LARP1, ADD3, and PHLPP2 [140]. This example highlights how the accumulation of RNA-seq data derived from clinical samples holds great potential to yield not only cancer-specific isoforms but also biomarkers of patient prognosis or response to therapy. In fact, certain other AS events have recently been reported to show prognostic value for ovarian, lung, pancreatic, prostate, and colorectal cancer patients [141,142,143,144,145]. For instance, in lung cancer, a genome-wide profiling identified various AS events significantly associated with patient survival [142], including EGFR, CD44, AR, RRAS2, MAPKAP1, and FGFR2. In CRC, two differently expressed AS events, namely, CSTF3-RI (intron retention) and CXCL12-AT (alternate terminator), were validated as independent prognostic indicators for both overall survival (OS) and disease-free survival [145]. Recently, the combination of high expression levels of COL6A3 E5-E6 junction and HKDC1 E1-E2 junction was for the first time associated with a better CRC patient OS [146]. Interestingly, it was previously reported that high gene expression of COL6A3 in stroma is linked to poor OS in CRC [147], while high expression of the *HKDC1* gene is related to poor OS in hepatocarcinoma [148], indicating that the biomarker value of some AS events is tumor-type or -stage specific. Additionally, some reports have focused on the identification of predictive biomarkers for drug response. For instance, Safikhani et al., based on a combined approach between pharmacological data and genome-wide transcriptomics, validated AS isoforms of IGF2BP2, NECTIN4, ITGB6, and KLHDC9 as predictive biomarkers for drug response to AZD6244, lapatinib, erlotinib, and paclitaxel, respectively [149]. As a whole, despite the potential biomarkers identified to date, they still require validation in independent patient cohorts and translation into clinical practice.

## 4. Therapeutic Strategies Targeting Alternative Splicing in Cancer

The identification of cancer-specific AS variations has guided the development of a multitude of promising therapeutic strategies. Actually, due to the different origins of AS dysregulation, previously discussed in Section 2.1, aberrant splicing programs in cancer can be targeted in diverse ways, as exemplified in Figure 2, including strategies such as blocking of protein kinases that post-translationally regulate splicing factors, disruption of signaling pathways regulating AS programs, use of oligonucleotides that modulate splicing factor recruitment to the pre-mRNA, targeting of protein isoforms derived from aberrant AS events, or targeting of the components of RNA spliceosome machinery. The latter has been discussed in detail elsewhere [29,150,151], and thus will not be addressed here.

### 4.1. Inhibiting Post-Translational Modifications of Splicing Factors or RNA Binding Proteins

Although several types of post-translational modifications of splicing factors were described, their phosphorylation has a key role, and the development of small-molecule inhibitors targeting protein kinases has emerged as a promising therapeutic strategy to reverse aberrant RNA splicing [152,153]. The two main targets of these molecules are the SR-rich protein-specific kinases (SRPKs) and the dual-specificity Cdc2-like kinases (CLKs), which primarily regulate pre-mRNA splicing by phosphorylating SR proteins, controlling both their nucleo-cytoplasmic shuttling and their interactions with the spliceosome [154]. Increased levels of these splicing regulatory protein kinases have been found in several types of cancers, which highlights the therapeutic potential of their pharmacological targeting. Particularly, upregulated expression of SRPK1 is frequently associated with an oncogenic activity in a variety of cancer types [155]. Accordingly, pharmacological inhibition of SRPK1 with the first-generation drug SRPIN340 induced splice switching of pro-angiogenic VEGFA165 to anti-angiogenic VEGFA165b (Figure 2A) in prostate cancer and leukemic cells [156,157]. Another study also showed that SRPIN340 significantly reduces tumor growth in metastatic melanoma in vivo via reduced expression of pro-angiogenic VEGF isoforms [158]. More recently, a covalent inhibitor of SPRK1 and SPRK2—SRPKIN-1—was developed, which efficiently reduced SR protein phosphorylation, promoted splice switching of VEGFA165 to VEGFA165b, and blocked neovascularization. This was achieved by local application in mice [159], but a corresponding benefit in tumor therapy remains to be demonstrated. The first CLK inhibitor to be discovered was the benzothiazole compound TG-003, which demonstrated selective potency toward CLK1, CLK2, and CLK4 and regulated splicing by reducing the phosphorylation of several SR proteins, including SRSF1 [160]. Two other compounds, leucettine L41 and T-025, were also identified as CLK inhibitors that modulate AS by the same mechanism of action [161,162]. Lastly, in a large-scale screening, a set of related compounds was identified, namely Cpd-1, Cpd-2, and Cpd-3, capable of targeting CLK1 and CLK2 and, to a lesser extent, SPRK1 and SRPK2 [163]. Despite the apparent success of these compounds in vitro, further studies are needed in order to improve their efficacy and narrow the window of off-target effects on splicing before moving to the clinical trial setting.

Another strategy to target splicing is exemplified by the use of sulfonamides, including E7820, indisulam, tasisulam, and chloroquinoxaline sulphonamide. These agents are known to show antitumor activity, and some of them have already been tested in clinical trials [164,165,166]. Later, it was confirmed that several sulphonamides interfere with splicing by promoting ubiquitin-mediated degradation of U2AF-related splicing factor RBM39 (also called CAPERα) via CRL4 E3 ubiquitin ligase complex [167,168]. For example, it was found that after indisulam treatment of cultured cancer cells, RBM39 degradation led to altered pre-mRNA splicing, including intron retention and exon skipping, in hundreds of genes [167]. However, it was also shown that RBM39 degradation is limited to certain cancer cells. Actually, mutations in RBM39, specifically in RRM2 domain, prevent its proteasomal degradation, thus conferring sulphonamide resistance. Sensitivity to these compounds also correlates with the expression levels of DCAF15 in hematopoietic and lymphoid lineages because the CUL4-DCAF15 complex regulates the ubiquitination and degradation of RBM39 [167].

### 4.2. Modulation of Signaling Pathways Regulating Alternative Splicing Events

The involvement of signaling pathways in the regulation of AS is well recognized, as referred to above. The mechanisms through which signal transduction pathways interfere directly or indirectly with splicing, typically involve regulation of either the cellular localization or the activation status of splicing-regulatory proteins. Therefore, modulation of signaling pathways represents a promising approach to target dysregulated AS. Importantly, despite the existence of a wide range of compounds able to target these pathways (reviewed in [169]), some of which have already been tested in clinical trials, they were not specifically developed to modulate AS. However, they proved to be valuable tools to further elucidate the mechanisms involved in the regulation of AS by oncogenic signaling pathways. For instance, the AKT inhibitor MK2206 was used to validate the results obtained in a study that aimed to unravel the differential regulation of the phosphoproteome by AKT isoforms [170]. Briefly, it was demonstrated that the specific RNA processing protein IWS1 is phosphorylated by AKT1 and AKT3 in lung cancer. IWS1 phosphorylation allows the recruitment of SETD2 to the RNA polymerase II complex. SETD2 trimethylates histone H3 during transcription, creating a docking site for PTBP1 splicing factor. In turn, PTBP1 promotes the skipping of exon IIIb in the fibroblast growth factor receptor 2 (*FGFR-2*) gene, shifting the balance of *FGFR-2* splicing from the IIIb to the IIIc isoform, which promotes cell proliferation, migration, and invasiveness in response to FGF-2. Moreover, in a work carried out in Ewing sarcoma cells, it was found that hnRNP M was strongly upregulated both at the mRNA and protein level upon inhibition of the PI3K/AKT/mTOR pathway with BEZ235, and located in the soluble nucleoplasmic fraction, where it modulated U1 snRNP recruitment to a 5′ splice site and triggered a splicing program contributing to drug resistance (Figure 2B) [171]. Overall, these types of inhibitory drugs have the ability to change splicing outcomes; however, it is crucial to invest in the development of compounds targeting specific abnormal AS events in order to limit the occurrence of undesired side effects.

### 4.3. Antisense Oligonucleotides

RNA-targeted therapies emerged in 1978 when Zamecnik and Stephenson described for the first time a chemically modified oligonucleotide that inhibited gene expression and viral replication of Rous sarcoma virus [172,173]. From there, antisense oligonucleotides (ASOs) have been extensively explored in the process of drug development and proven to be a useful alternative approach for the target-specific treatment of splicing-related human diseases, including cancer. Briefly, ASOs are synthetic molecules consisting of short single-stranded nucleic-acid sequences, generally 15–25 nucleotides in length, that specifically bind through Watson–Crick base-pairing to complementary pre-mRNA sequences [174]. RNA-targeted therapies are already used in the clinic and numerous clinical trials with therapeutic ASOs are currently underway [175,176,177].

The antisense therapies can be subdivided into two groups according to their downstream mechanisms of action and functional outcomes. The majority of ASOs are designed to promote the cleavage of targeted mRNA by endogenous cellular nucleases, such as RNase H, which recognizes double-stranded RNA:DNA hybrids and subsequently degrades the disease-causing gene product. A different strategy aims to interfere with the access of the splicing machinery to the regulatory sequences in the pre-mRNA instead of causing the transcript degradation. So-called splice-switching antisense oligonucleotides (SSOs) are designed to compete with and sterically block the binding of certain splicing factors to their specific sites in the pre-mRNA, which in turn changes exon recognition by the spliceosome [178]. As such, this strategy intends to specifically shift the splicing pattern of a targeted pre-mRNA transcript, favoring the production of one of the splicing variants with potential therapeutic benefits. To date, two SSOs were approved by the US Food and Drug Administration (FDA), Eteplirsen and Nusinersen, for the clinical treatment of the genetic diseases Duchenne muscular dystrophy and spinal muscular atrophy, respectively [179,180]. Eteplirsen hybridizes at exon 51 of the *DMD* pre-mRNA (which encodes the dystrophin protein), sterically blocking the recognition of this exon by the spliceosome and thereby promoting the skipping of exon 51 to correct the disease-causing frameshift mutation and generating a shorter but functional variant of the protein (Figure 2C) [181]. In a distinct way, Nusinersen binds to an intronic region upstream of exon 7 in the *SMN2* pre-mRNA that encodes the survival motor neuron protein [180]. The binding blocks recruitment of an inhibitory splicing factor that would normally impede the recognition of exon 7 by the spliceosome, thus enhancing the inclusion of the formerly missing exon 7 of *SMN2*, and the subsequent production of a fully functional protein that is absent in patients with spinal muscular atrophy. Although the application of SSOs in anticancer therapy is still under evaluation, the modulation of RNA splicing of cancer-related genes has been successfully achieved in various pre-clinical cancer models. One of the most used strategies targets the *BCL2L1* gene that is alternatively spliced, originating either anti-apoptotic Bcl-xL or pro-apoptotic Bcl-xS proteins. Thus, in order to abolish the high expression levels of Bcl-xL reported in many cancers, Bcl-x SSOs were designed to induce a splicing switch, favoring the production of the pro-apoptotic isoform Bcl-xS. It was shown in vitro that treatment with these SSOs shifted splicing from Bcl-xL to Bcl-xS in various cancer cell lines [182]. Moreover, it was found that Bcl-xS proteins induced by the SSOs sensitized the cancer cells to treatment with chemotherapeutic agents or ultraviolet (UV) radiation [70]. Additionally, the antitumor activity of SSOs was also demonstrated in vivo in a mouse model of melanoma lung metastases where the systemic administration of Bcl-x SSO using a lipid nanoparticle redirected Bcl-x splicing and led to a significant reduction in tumor burden in treated mice [183]. Another important antitumor target is the hnRNP-regulated splicing of the *PKM* gene [184], which is a critical player in the regulation of glucose metabolism by producing either the PKM1 isoform (that stimulates oxidative phosphorylation) or the PKM2 isoform (that promotes aerobic glycolysis, a metabolic shift also recognized as the Warburg effect). PKM2 is frequently upregulated in cancer cell lines and various tumor types, including CRC, and SSOs used to switch the expression back to PKM1 induced apoptosis [185]. Further examples of SSO-mediated splicing modulation of other genes, including *BCL2L11*, *BRCA1*, *ERBB2*, *MDM4*, *MKNK2*, and *STAT3*, were recently reviewed in [150]. A related SSO approach is the design of decoy oligonucleotides composed of the RNA motif recognized by a given splicing factor, which can downmodulate its splicing activity. This could be a promising therapeutic approach whenever a splicing factor is either overexpressed or hyperactived in cancer cells [186].

### 4.4. Targeting the Alternative Protein Isoform

The presence of specific AS variant proteins in tumor cells suggest them as potential therapeutic targets. Some variants may result in the translation of immunogenic neoantigens, either as a result of frameshifts or re-expression of developmental variants. As such, some strategies have been developed to target cancer-specific isoforms by immunotherapies. One of the most explored therapeutic targets are EGFR variants de4 and vIII. Although in GBM and other cancers EGFRvIII results from a genomic deletion of exons 2–7 [187], an AS variant with skipping exon 4 leads to a comparable phenotype in other tumors: a lack of amino acids in the extracellular ligand-binding domain, resulting in a constitutively active variant able to stimulate downstream signaling in a ligand-independent manner. Several studies have supported the oncogenic role of these EGFR variants and their association with a poor prognosis [103,104,105,124,125,188,189,190,191,192]. Being tumor-specific cell surface molecules, these receptor variants were successfully targeted by therapeutic antibodies [104,189]. Notably, in 2015, the vaccine rindopepimut (also known as CDX-110), consisting of an EGFRvIII-specific peptide conjugated to keyhole limpet haemocyanin, was approved by FDA for the treatment of GBM. Actually, the results obtained in phase I and II clinical trials showed that the treatment with rindopepimut increases both OS and progression-free survival of GBM patients expressing EGFRvIII [189]. Additionally, the role of cell adhesion molecule CD44 and its isoforms containing the exon v6 have been broadly implicated in the metastatic tumor process, and as such they have also been explored as targets for anticancer therapy [193]. One of the most recognized anti-CD44v6 therapy consists of using bivatuzumab, a humanized IgG1 monoclonal antibody labelled with rhenium-186 (Figure 2D). Particularly in phase I clinical trials for patients with the head and neck squamous cell carcinoma (HNSCC), a tissue that expresses high amounts of CD44v6 antigen, bivatuzumab showed promising antitumor effects with consistent stable disease at higher radioactive dose levels and with low toxicity [194,195]. Based on these results, a novel strategy comprising the coupling of bivatuzumab with a non-radioactive cytotoxic drug, mertansine, was developed [196]. Interestingly, in phase I clinical trials, the intravenous injection of bivatuzumab mertansine in adult patients with recurrent or metastatic HNSCC induced a partial response in three of the 30 patients tested, which presented a stabilization of the disease and regression of tumors [197]. Despite the promising results, the toxic side effects observed in the skin led to the discontinuation of the clinical trials with bivatuzumab mertansine. Another example is the tight junction molecule claudin-18 isoform 2 (CLDN18.2). In 2008, CLDN18.2 was identified as a highly selective cell lineage marker, whose expression in normal tissues is restricted to differentiated epithelial cells of the gastric mucosa, being absent from the gastric stem cell zone [198]. Additionally, it was also reported that CLDN18.2 is expressed in a significant proportion of primary gastric cancers and their metastases. Since CLDN18.2 exposes extracellular loops available for antibody binding, a targeted therapy based on the monoclonal antibody IMAB362 (claudiximab) was developed [199]. According to the promising results obtained in previous clinical trials, a phase III global study of IMAB362 plus FOLFOX versus FOLFOX plus placebo as first-line treatment was initiated in 2018 in gastric cancer patients (NCT03504397).

Besides these immunotherapeutic approaches, protein–protein interaction inhibitors could become a promising precision-medicine approach for targeting AS-derived protein isoforms. Many AS variants generate proteins following exon inclusion or intron retention and can contain extra protein domains that participate in protein–protein interactions involved in their downstream function. Small-molecule drugs that compete with these interactions are being developed [200,201]. For example, the BCL-2-selective inhibitor ABT-199 competes for anti-apoptotic interaction with BAK/BAD proteins [202], and inhibitors of the MDM2–p53 complex can restore p53 function in cancerous cells, leading to their growth arrest and apoptosis [203].

## 5. Concluding Remarks and Future Perspectives

From a large body of experimental data, it has become increasingly clear that AS is tightly associated with human health and disease [3]. However, despite AS being the major driver of biological diversity and playing a role in every hallmark of cancer, it was neglected for a long time in the profiling of tumor characteristics and overlooked as a source of new biomarkers and therapeutic targets for drug development. Nevertheless, with the emergence of advanced sequencing technologies that provide a landscape of AS at a genome-wide level, additional insights into the splicing programs were achieved [6,7,8,9]. Remarkably, these studies have contributed in a decisive way to the identification of aberrant AS events in cancer development and progression, a prerequisite for the identification of potential biomarkers and development of new therapeutic strategies towards cancer precision medicine. Despite the described progress, the tumor-specific splicing alterations are far from being characterized and further efforts are needed to provide a comprehensive view of splicing regulation and of its dysregulation in cancer.

Another important aspect that has emerged from advanced sequencing technology is the need to move our understanding from individual AS variants to the overall pattern of splicing changes in tumors. Any change in activity or localization of a splicing factor will potentially trigger a plethora of AS decisions in many different genes. Thus, AS signature profiles or patterns may represent more meaningful biomarkers. Regarding AS-targeting drug development, existing small-molecule compounds do mostly interfere with early spliceosome assembly or post-translational modification of SR proteins, but lack efficient antitumor activity. As such, the recent efforts focused on the targeting of pathological RNA isoforms or tumor-specific protein variants represent the most promising attempts to develop more effective drug candidates. Unfortunately, these targeted anticancer therapies based on AS are still far from reaching the clinic. To address this issue, it is a priority to reveal in more detail how altered AS actually drives tumorigenesis, and how it is connected to altered genotypes or signaling pathways that characterize tumor phenotypes.

## Figures and Tables

**Figure 1 ijms-21-09032-f001:**
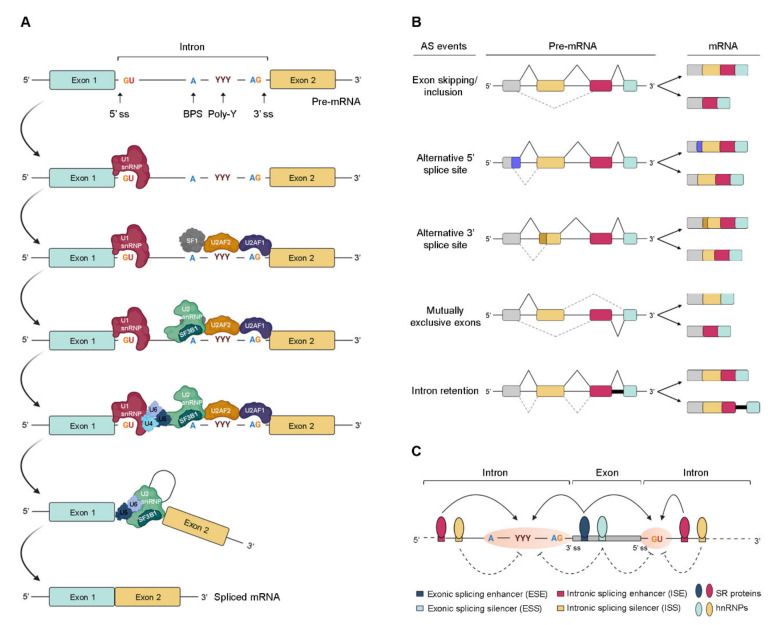
Regulation of pre-mRNA splicing. (**A**) Stepwise assembly of spliceosome on the pre-mRNA and catalysis of the splicing reaction to generate mature spliced mRNA. (**B**) Schematic representation of the most common alternative splicing AS events. The grey, yellow, red, and blue boxes represent different exons. The solid black and dotted grey lines indicate distinct splicing events. (**C**) Complex interplay between cis- and trans-acting factors in the regulation of AS. RNA-binding motif (RBM) proteins, serine/arginine-rich (SR) proteins, and heterogeneous (hn) ribonucleoproteins (hnRNPs) bind to exonic or intronic regulatory elements to promote or prevent the recognition of either 3′ or 5′ splice sites (ss) by the small nuclear (sn) RNPs (snRNPs) and splicing factors. The solid and dotted black arrows represent binding stimulation and inhibition, respectively; (ss—splice sites; BPS—branch point site; poly-Y—polypyrimidine tract; pre-mRNA—precursor messenger RNA; snRNPs—small nuclear ribonucleoprotein particle; SF1—splicing factor 1; U2AF—U2 snRNP auxiliary factor).

**Figure 2 ijms-21-09032-f002:**
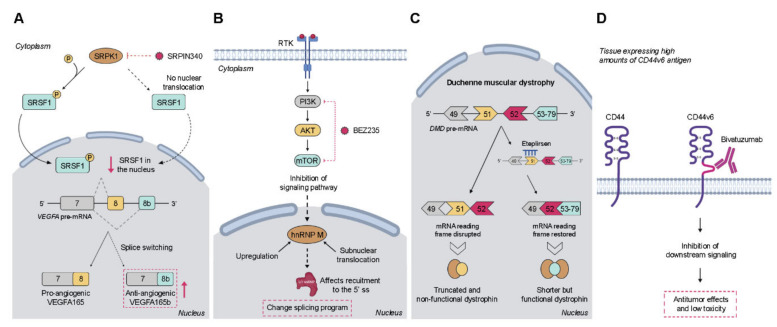
Examples of therapeutic strategies targeting alternative splicing (AS). (**A**) Targeting of protein kinases by small molecules to inhibit the post-translational phosphorylation of splicing factors. (**B**) Inhibition of signaling pathways by small molecules. (**C**) Splice-switching antisense oligonucleotides. (**D**) Targeting of cancer-specific isoforms by therapeutic monoclonal antibodies; (pre-mRNA—precursor messenger RNA; ss—splice site; RTK—receptor tyrosine kinase; hnRNP—heterogeneous nuclear ribonucleoprotein; *DMD*—dystrophin gene).

**Table 1 ijms-21-09032-t001:** Tumor-associated AS variants and the respective cancer-promoting process.

Gene	Splicing Event	Biological Function	Cancer Types	References
*BCL2L1*	5′ alternative splice site usage in exon 2	Bcl-xL inhibits apoptosis	Lymphoma, glioma, breast, prostate, and liver cancer	[67,68,69,70,71]
*MKNK2*	Skipping of exon 14a and inclusion of exon 14b	MNK2b acts p38-MAPK-independent and promotes cell growth	Breast, colon, and lung cancer	[44,72,73]
*PKM*	Skipping of exon 9 and inclusion of exon 10	PKM2 stimulates aerobic glycolysis	Ovarian, gastric, liver, and colon cancer	[74,75,76,77]
*MST1R (RON)*	Skipping of exon 11	RONΔex11 induces cell motility and invasion	Colon, ovarian, brain, lung, and gastric cancer	[78,79,80,81,82]
*RPS6KB1*	Inclusion of three cassette exons 6a, 6b, and 6c with a PTC in exon 6c	RPS6KB1-2 promotes cell proliferation and tumor growth	Breast and lung cancer	[83,84]
*CCND1*	5′ alternative splice site usage in exon 4 introduces a PTC	Cyclin D1b induces invasion and metastasis	Breast, lung, and prostate cancer	[85,86,87]
VEGFA	Alternative 3′ splice site in exon 8	VEGFA165 has pro-angiogenic activity	Colon, prostate, renal, and skin cancer	[88,89,90,91]
*CEACAM1*	Inclusion of exon 7	CEACAM1-L accelerates metastasis progression	Colon cancer and metastatic melanoma	[92,93]
*CD44*	Inclusion of variable exon 6	CD44-v6 induces migration and expression of mesenchymal markers	Colon cancer	[94,95,96]
*RAC1*	Inclusion of exon 3b	RAC1B increases cell survival and transformation	Colon, pancreas, thyroid, breast, and lung cancer	[63,97,98,99,100,101,102]
*EGFR*	Skipping of exon 4	de4-EGFR promotes malignant transformation as constitutively active receptor variant	Glioma, prostate, and ovarian cancer	[103,104,105]
*KLF6*	5′ alternative splice site usage in exon 2	KLF6-SV1 lacks nuclear localization and contributes to mesenchymal phenotype	Breast, lung, pancreatic, prostate, and liver cancer	[106,107,108,109,110]
*CTTN*	Inclusion of exon 11	Cortactin isoform-a increases cell migration	Colorectal cancer	[111]
*FAK*	Deletion of exon 26	The −26-exon FAK isoform is caspase-resistant and inhibits apoptosis	Breast cancer	[112]

The listed genes are B-cell CLL/lymphoma 2-like 1 (*BCL2L1*), MAPK interacting serine/threonine kinase 2 (*MKNK2*), pyruvate kinase M (*PKM*), macrophage stimulating 1 receptor (*MST1R*), ribosomal protein S6 kinase B1 (*RPS6KB1*), cyclin D1 (*CCND1*), vascular endothelial growth factor A (*VEGFA*), CEA cell adhesion molecule 1 (*CEACAM1*), clusters of differentiation 44 (*CD44*), ras-related C3 botulinum toxin substrate 1 (*RAC1*), epidermal growth factor receptor (*EGFR*), Krüppel-like factor 6 (*KLF6*), cortactin (*CTTN*), and focal adhesion kinase (*FAK*); (PTC—premature termination codon).

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
