# Peer review of "Alternative Splicing: Expanding the Landscape of Cancer Biomarkers and Therapeutics"

_ijms, 2020, doi:10.3390/ijms21239032_

Round 1
Reviewer 1 Report
In this review article Bessa and collaborators discuss the potential value of dysregulated splicing signatures in diagnosis and its exploitation in innovative therapeutic approaches
This is a timely review, very well written and provides a complete overview of the this topic.
Minor points
I found funny that for three times authors misspelled splicing/spliceosome
1- page 2: splcicing
2- page 4: spliceossome
3- page 7: splicingevents
Please define what OS stands for
Author Response
We thank the reviewer for the positive comments and have corrected all indicated spelling errors.
Reviewer 2 Report
I thoroughly enjoyed reading this review article. The authors started with a broad introduction on alternative splicing and expanded on its role in cancer. Both the table and figures are very detailed and well made. Overall, a great job! I have some minor changes as follows.
I think an introduction to cancer is required before describing the role of alternative splicing in cancer. This can be brief.
Line 56 correct splicing spelling
Line 58 remove extra space
Line 202 correct interestingly spelling
Line 337 correct phosphorylation spelling
Line 468 correct adhesion spelling
Author Response
We thank the reviewer for the positive comments and have corrected all indicated spelling errors.We also followed the suggestion and included a small introductory context about cancer at the beginning of section 2.1. Dysregulation of alternative splicing in cancer, as follows.
2.1. Dysregulation of alternative splicing in cancer
Cancer mainly evolves through successive genetic alterations and genomic dysregulation, but is also affected by thetumor microenvironment. These renderoncogenes constitutively active, and inactivate tumour-suppressor genes.As a result, cancer cells acquire specific abilities during tumordevelopment, including self-sufficiency in growth signals, insensitivity to growth inhibitory signals, evasion of apoptosis, limitless replicative potential, sustained angiogenesis, and tissue invasion and metastasis [19]. These processes can also be dysregulated by AS, which in turn can generate variant proteins with altered physiological function[3]. Particularly, a recent systematic study performed by Kahles and colleagues reported that AS events are more frequent in cancer tissues compared to(...)